# Introgression of the Powdery Mildew Resistance Genes *Pm60* and *Pm60b* from *Triticum urartu* to Common Wheat Using Durum as a ‘Bridge’

**DOI:** 10.3390/pathogens11010025

**Published:** 2021-12-26

**Authors:** Qiang Zhang, Yinghui Li, Yiwen Li, Tzion Fahima, Qianhua Shen, Chaojie Xie

**Affiliations:** 1State Key Laboratory for Agrobiotechnology, Key Laboratory of Crop Heterosis and Utilization (MOE), Key Laboratory of Crop Genetic Improvement, College of Agronomy and Biotechnology, China Agricultural University, Beijing 100193, China; zhang2021qiang2021@163.com; 2State Key Laboratory of Plant Cell and Chromosome Engineering, Institute of Genetics and Developmental Biology, Innovation Academy for Seed Design, Chinese Academy of Sciences, Beijing 100101, China; ywli@genetics.ac.cn; 3Institute of Evolution, University of Haifa, Mt. Carmel, Haifa 3498838, Israel; tfahima@univ.haifa.ac.il; 4CAS Center for Excellence in Biotic Interactions, University of Chinese Academy of Sciences, Beijing 100049, China

**Keywords:** wheat powdery mildew, *Triticum urartu*, *Pm60*, recombinant types, durum as a bridge, introgression lines

## Abstract

Powdery mildew, caused by the fungus *Blumeria graminis* f. sp. *tritici* (*Bgt*), has limited wheat yields in many major wheat-production areas across the world. Introducing resistance genes from wild relatives into cultivated wheat can enrich the genetic resources for disease resistance breeding. The powdery mildew resistance gene *Pm60* was first identified in diploid wild wheat *Triticum urartu* (*T. urartu*). In this study, we used durum as a ‘bridge’ approach to transfer *Pm60* and *Pm60b* into hexaploid common wheat. Synthetic hexaploid wheat (SHW, AABBA^u^A^u^), developed by crossing *T. urartu* (A^u^A^u^) with durum (AABB), was used for crossing and backcrossing with common wheat. The *Pm60* alleles were tracked by molecular markers and the resistance to powdery mildew. From BC_1_F_1_ backcross populations, eight recombinant types were identified based on five *Pm60*-flanking markers, which indicated different sizes of the introgressed chromosome segments from *T. urartu*. Moreover, we have selected two resistance-harboring introgression lines with high self-fertility, which could be easily used in wheat breeding system. Our results showed that the durum was an excellent ‘bridge’ for introducing the target gene from diploid *T. urartu* into the hexaploid cultivated wheat. Moreover, these introgression lines could be deployed in wheat resistance breeding programs, together with the assistance of the molecular markers for *Pm60* alleles.

## 1. Introduction

*Blumeria graminis* f. sp. *tritici* (*Bgt*) is the disease-causing fungus of wheat powdery mildew, which is one of the most destructive diseases in many major wheat production areas. The disease could be well controlled by genetic resistance in the host; however, throughout history, the pathogen has overcome widely deployed host resistance genes rather rapidly [1]. Until now, more than 80 designated powdery mildew resistance genes (*Pm*) have been identified in wheat [2]. A significant number of these genes are derived from wild relatives of common wheat, which have chromosomes homologous to that of common wheat. For example, *PmG16*, *Pm41,* and *MlNFS10* are derived from *Triticum turgidum* L. var. *dicoccoides* [3,4,5], *Pm13* and *Pm66* from *Aegilops longissima* [6,7], *Pm12* and *Pm53* from *Ae*. *speltoides* [8,9], *Pm21* from *Dasypyrum villosum* [10], *Pm60* from *T. urartu* [11], and *Pm8* from *Secale cereale* [12].

The wild diploid wheat, *T*. *urartu* (2n = 2x = 14; genome A^u^A^u^), is the A-genome donor of tetraploid wheat *T. turgidum* subsp. *durum* (2n = 4x = 28; genome AABB) and hexaploid common wheat *T. aestivum* (2n = 6x = 42; genome AABBDD) [13]. *T*. *urartu* has extensive genetic diversity for many traits, including disease resistance and phenological and morphological characteristics [14]. The natural populations of *T. urartu* are extremely diverse, with adaptive traits that have accumulated over evolutionary time, which allow them to grow under a wide range of environmental conditions [15]. Some *T. urartu* materials have good resistance to wheat diseases, including powdery mildew, stripe rust, and stem rust [16,17,18], which makes *T. urartu* a genetic resource for wheat improvement to enrich the gene pool of wheat [19]. Because of its homology to the A subgenome of durum wheat and common wheat, genes with superior traits in *T. urartu* can be transferred to tetraploid and hexaploid cultivated wheat through direct hybridization and gene introgression [20,21].

Wide crossing has been an important approach for crop improvement, in which segments containing favorable genes from wild relatives can be introduced into existing crop species through chromosomal recombination [22]. However, the rate of success is very low when directly crossing *T. urartu* or other wild wheat relatives with hexaploid wheat [23]. In some cases, the use of durum as a ‘bridge’ approach could be helpful in transferring the genes from wild wheat into common wheat [24]. There are many examples of the utilizations of this approach for the introgression of wild-wheat-derived resistance genes into common wheat (e.g., *Sr60*, *Yr15*, *Yr36*, and *PmG16* [3,24,25]; reviewed by Klymiuk [24]). *T. urartu* is the donor of the A genome of *T. durum*, and durum shares the same A and B genomes with *T. aestivum*. Therefore, durum wheat might be suitable also as a ‘bridge’ between *T. urartu* and common wheat for the introduction of exogenous genes.

To date, only one powdery mildew resistance gene, *Pm60*, has been cloned from *T. urartu* [11]. *Pm60* encodes a protein with a nucleotide-binding site (NBS) and leucine-rich repeat (LRR) domain. Three functional resistance alleles were identified on the *Pm60* locus (*Pm60*, *Pm60a*, and *Pm60b*), distinguishable by an 80 amino acid deletion (*Pm60a*) or insertion (*Pm60b*) in the LRR region as compared with *Pm60* [11].

In this study, we introgressed *Pm60* and *Pm60b* from *T. urartu* into common wheat using the durum wheat line Mo75 as a ‘bridge’. During the introduction, *Pm60* and *Pm60b* alleles were selected in the progenies via functional molecular markers, and verified according to the resistance phenotype to powdery mildew. Based on graphical genotype analysis, eight different recombinant introgression types were identified that carry different lengths of chromosome segments derived from *T. urartu* with one of the *Pm60* alternative alleles. At the same time, we selected two resistant lines with high self-fertility. These resistant introgression lines provide excellent materials for future wheat powdery mildew resistance breeding.

## 2. Materials and Methods

### 2.1. Plant Material

*T. urartu* accessions CITR17664 (Baal Bek-Bashari, Lebanon, *Pm60*), PI428215 (Mardin, Turkey, *Pm60b*) and PI428315 (Baal Bek-Bashari, Lebanon, *Pm60b*) carrying the powdery mildew resistance genes *Pm60* and *Pm60b* were used as the *Pm60* donor lines [26]. The durum wheat line Mo75, with high susceptibility to *Bgt* isolate E09, was used as a bridge for the introgression of *Pm60* alleles into common wheat (Figure 1). Common wheat lines Fielder, Xuezao, and Shiluan 02–1, which are susceptible to *Bgt* isolate E09, were used as recurrent parents.

### 2.2. Introgression of Disease Resistance Genes

To introduce the *Pm60* and *Pm60b* from *T. urartu* into common wheat, a hybridization strategy was devised (Figure 1). The susceptible durum wheat line Mo75 was used as the female to cross with the *T. urartu* accessions containing *Pm60* and *Pm60b*, and the F_1_ seeds were harvested. The hybrid seeds were planted in a greenhouse and grown to young plants (ABA^u^). These plants were removed from the soil at the five- or six-leaf stage for artificial chromosome doubling (AABBA^u^A^u^). After the soil was gently washed off the roots, the plants were submerged in a mixture of 2.5mM colchicine (0.1%, *w*/*v*) and 2% dimethyl sulfoxide (DMSO) and kept in darkness for 24 h at 7 °C–10 °C in the growth chamber [27]. After treatment, all plants were thoroughly washed and transplanted in the greenhouse to achieve chromosome doubling (capable of self-crossing), thus creating synthetic hexaploid wheat with the genome of AABBA^u^A^u^. The succeeded synthetic hexaploid wheat (SHW) plants could be self-fertile, or were otherwise sterile. The SHW plants were used as the parent to cross with the susceptible common wheat line Xuezao, and the hybrid progeny were evaluated for powdery mildew resistance in the greenhouse. To introduce the powdery mildew resistance genes into more common wheat backgrounds, the resistant hybrid plants were used to cross and backcross with several susceptible wheat lines Fielder, Xuezao, and Shiluan 02–1 in the next crossing process.

### 2.3. Bgt Inoculation and Disease Assessment

The *Bgt* isolate E09, a prevailing pathotype in the Beijing area, is virulent to the resistance genes *Pm1a*, *Pm3a*, *Pm3c*, *Pm5a*, *Pm7*, *Pm8*, *Pm17*, and *Pm19* [28], and was used in this study. The experimental material was cultivated in a growth chamber with a relative humidity of 75%, a diurnal temperature of 26 °C/20 °C, a photoperiod of 14/10 h (light/dark), and a light intensity of 3000 lx. The *Bgt* isolate E09 was maintained on the susceptible wheat line Xuezao by transferring spores to new plants each week. Inoculations were performed by dusting or brushing conidia from neighboring sporulating susceptible seedlings onto the two-week-old wheat seedlings. Approximately 10–14 days after inoculation, when susceptible control plants were fully infected, the infection types (IT) of the tested plants were scored using the 0–4 scale, where 0 indicated immune with no visible symptoms, 0; for necrotic flecks we noted a hypersensitive response (HR), 1 for necrosis with little sporulation, 2 for necrosis with moderate sporulation, 3 for moderate-to-high sporulation without necrosis, and 4 for full sporulation without necrosis [29]. ITs 0–2 were considered resistant and ITs 3–4 were susceptible.

### 2.4. Tracing of Pm60 Allele by Molecular Markers

Total genomic DNA was extracted from young seedling leaves by means of the cetyltrimethylammonium bromide (CTAB) method [30]. DNA samples were quantified using a NanoDrop One spectrophotometer instrument (Nanodrop Technologies, Wilmington, DE, USA) and diluted to a concentration of 30 ng/µL.

The *Pm60* alleles were tested via amplification with *M-Pm60* primers [11]. The flanking markers of *Pm60* used in the experiments were all from Zou et al. [11] (Table 1). Polymerase chain reaction (PCR) amplification was performed in a 20 µL reaction volume, containing 10 µL of 2 × Taq PCR StarMix (Genstar, Beijing, China) and loading dye, 4 µL of 30 ng/mL DNA, 2 µL of primers (mixture of forward and reverse primers, 2 mM), and 4 µL of ddH_2_O. PCR amplification was performed using the marker M-*Pm60* with a protocol of 5 min at 94 °C; 35 s at 94 °C, 35 s at 55 °C, 1 min 40 s at 72 °C, and termination after 10 min of extension at 72 °C for a total of 35 cycles. For the flanking markers, the primer extension time was 30 s at 72 °C and the other PCR conditions were the same as those described above. PCR products were separated in 1% agarose gels or via 10% non-denaturing polyacrylamide gel electrophoresis (acrylamide: bisacrylamide = 39:1), and gels were visualized with silver nitrate staining [31].

### 2.5. Analysis of the Introgressed Fragments Flanking Pm60 Locus

Analysis of the introgression fragments flanking *Pm60* locus in terms of the number and classes of genes and proteins was performed using annotated information of *T. urartu* (G1812, Tu2.0) and Chinese Spring (RefSeq 1.1) from the Triticeae Multi-omics Center database (http://202.194.139.32/getfasta/index.html) (accessed on 3 March 2021).

## 3. Results

### 3.1. Synthesizing Hexaploid AABBA^u^A^u^ Wheat by Crossing T. urartu (A^u^A^u^) with Durum (AABB)

In the previous report, Zhao et al. [26] found that *T. urartu* accessions CITR 17664 contained *Pm60*, PI 428215, and PI 428315 contained *Pm60b*. To introduce *Pm60* and *Pm60b* into common wheat, the susceptible durum wheat line Mo75 was used as a ‘bridge’. Firstly, we crossed those three *T. urartu* accessions (CITR17664, PI428215, PI428315; A^u^A^u^) with the female parent Mo75 (AABB), and obtained 3–10 F_1_ seeds for each cross. The hybrid plants were treated with colchicine for artificial chromosome doubling. Finally, we got 1–3 plants with high self-fertility from each cross, suggesting that the chromosomes in these plants were successfully doubled. Compared with the durum wheat line Mo75, synthetic hexaploid wheat (SHW, AABBA^u^A^u^) showed some new morphological traits, such as spike brittleness, larger seeds, and longer spikes (Figure 2), which were probably contributed by *T. urartu*. A powdery mildew test indicated that these SHWs were highly resistant (Appendix A; IT = 0–0;) to *Bgt* isolate E09 at all stages, suggesting that the resistance of *Pm60* alleles from diploid *T. urartu* remained effective and was not suppressed in these hexaploid SHW.

### 3.2. Introgression of Pm60 Alleles into Common Wheat 

To introduce *Pm60* alleles into the common wheat cultivars, we crossed the SHW accessions with the susceptible common wheat line Xuezao. As expected, all the F_1_ plants were immune to *Bgt* isolate E09 (Table 2). Since the F_1_ plants with a genome of AABBDA^u^ were sterile, these plants were (back)crossed with several susceptible wheat lines, such as Xuezao, Fielder, and Shiluan 02–1. We obtained six BC_1_F_1_ populations using different male parents (Z1, Z2, Z5, Z7, Z10, and Z14, Table 2). After testing the resistance to *Bgt* isolate E09, we found that four populations (Z1, Z5, Z7 and Z10) met the 1:1 segregation ratio, except for Z2 and Z14 (Table 2). These results indicated that there is one dominant resistance gene segregating (ratio = 1:1) in these populations.

To verify the presence of *Pm60* and *Pm60b* in these progenies, we adopted the reported marker *M-Pm60* (Table 1) [11], a PCR-based functional marker of *Pm60* that can distinguish between the three alleles (*Pm60*, *Pm60a*, and *Pm60b*) based on the size of the amplification product [11]. Screening of 194 plants of these populations using the marker *M-Pm60* confirmed the presence of either the resistance gene *Pm60* (PCR product size: 1551 bp) or the *Pm60b* allele (PCR product: 1791 bp) in all the powdery-mildew-resistant plants, which were absent in all of the susceptible plants (Figure 3A,B). Therefore, by combining the phenotyping and genotyping results we were able to confirm that the functional *Pm60* alleles were successfully transferred from the diploid *T. urartu* donor lines into hexaploid wheat introgression lines.

### 3.3. Graphical Genotypes Analysis of the Introgressed Chromosome Segments 

The introgressions of favorable alien genes are often accompanied by the simultaneous transfer of different sizes of chromosomal fragments alongside the target genes [32,33]. Since *T. urartu* is the A genome donor for common wheat, its chromosomes could freely pair and recombine with the A sub-genome of common wheat. To analyze the sizes of the introgressed chromosome fragments harboring the *Pm60* locus in the common wheat genome background, we screened the BC_1_F_1_ progenies (focusing on the resistant individuals) using the *Pm60* flanking markers on the long arm of chromosome 7A (Appendix A). Among these, a total of 16 recombinant events were identified, which could be separated into eight types by the recombination sites based on the genotypes of five *Pm60*-flanking markers (Figure 4 and Appendix A). There were ten recombinants identified in Xuezao, three in the Shiluan 02–1, and three in Fielder. The markers further away from the target gene were more likely to be recombined (Figure 4 and Appendix A). Based on the annotation information of the *T. urartu* material G1812, some introduced chromosome segments were about 18.12–35.59 Mb of the long arm of *T. urartu* chromosome 7A in different recombinants, harboring 411–511 genes (195–328 protein species) from *T. urartu* (Figure 4). Interestingly, we identified an individual Z1–3 containing double recombination with *Pm60b* flanking the chromosome segments from Xuezao and Fielder (Figure 4 and Appendix A). These results indicated that chromosome 7A of *T. urartu* was fully homologous to chromosome 7 of common wheat and the size of the introgressive linkage fragments alongside the *Pm60* locus could readily be reduced by recombination.

Among the six BC_1_F_1_ introgressive populations, we found two recombinant progenies (Z1–4 and Z1–10, which were resistant and contained *Pm60b*) showing high self-fertility with 83.3% and 84.6% fertility rates, respectively (Figure 5). These two materials can be continuously selected to obtain homozygous resistant lines in the next generations and can be used for wheat breeding. However, the other recombinant progenies showed low self-fertility rates which were close to sterile. All of those selected recombinant progenies have been backcrossed with bread wheat and will develop more introgression lines in the future.

## 4. Discussion 

The wild relatives of wheat are a tremendous treasure chest of genetic variation for wheat improvement [34]. The incorporation of favorable traits from wild relatives into common wheat is regarded as a valuable way for breeders to increase the genetic diversity of wheat [35]. Molecular markers are now widely used for gene targeting, marker-assisted selection and other genetic studies, allowing tracking of the target gene through different generations of a breeding program, although independently of environmental conditions and growth stage [36]. In this study, the functional marker *M-Pm60* for *Pm60* was applied to test and track *Pm60* alleles in the introgression of the target gene, and flanking markers linked to the *Pm60* locus were used for the analysis of the size of the introduced fragments (Table 1 and Figure 3B). Taking advantage of those markers, we successfully selected the introgression lines containing *Pm60* alleles with different chromosome segments introgressed from *T. urartu* (Figure 4 and Appendix A).

*Pm60* alleles have been identified in approximately 34.8% (67 out of 227) of *T. urartu* natural populations and approximately 25.2% (58 out of 230) wild emmer wheat accessions [3,26]. These results suggest that the *Pm60* locus is one of the important *Bgt* resistance loci in the natural populations of *T. urartu* and wild emmer wheat. The different *Pm60* alleles might be resistant to different *Bgt* isolates. *TdPm60* has been transformed from wild emmer wheat into common wheat [3,37]. Since the resistances of *Pm60* and *Pm60b* genes had never been used in common wheat before, in this study, we introduced the *Pm60* and *Pm60b* genes into common wheat and found that their resistances to *Bgt* E09 were maintained in the hexaploid background (Figure 3A). The obtained introgression lines of *Pm60* alleles will enrich the *Pm* gene pool for wheat breeding.

Synthetic hexaploid wheats (SHWs) have been successfully used to combine the AB genome of tetraploid wheat and the D genome of *Ae. tauschii* to create synthesized AABBDD wheat [38]. Disease resistance genes of *Ae. tauschii* have been introduced into common wheat by means of this approach, such as the rust resistance genes *Lr21* and *Sr33* [39,40]. In our study, SHWs with the genome of AABBA^u^A^u^ were developed through the chromosome doubling of the hybrids of *T. durum* (AABB) and *T. urartu* (A^u^A^u^). Since SHW lines directly combined the whole genome of diploid wild relatives, they showed many undesirable traits, such as late maturity, taller plants, and difficulty in threshing (Figure 2). Therefore, SHWs could be used as a ‘bridge’ to introduce desirable genes into common wheat to develop superior derivatives (synthetic backcross lines, SBLs) or synthetic derivatives (SYN-DER) with the unfavorable traits eliminated. For example, four SHW derivatives were released during 2003–2005 in Sichuan, China. Among them, Chuanmai 42 has proven to be both a high-yielding variety and an excellent parent, from which 12 varieties have been bred [41]. In this study, we found that the AABBA^u^A^u^ SHWs were easy to cross with common wheat and could be used for the introduction of genes from the A^u^ genome into common wheat. By crossing SHWs with common wheat and continuously backcrossing with common wheat, we introduced *Pm60* alleles into different wheat varieties (Fielder, Xuezao, and Shiluan 02–1) and created highly resistant and fertile derivative lines (Figure 5). However, the SHW also showed unfavorable traits such as spike brittleness and wizened seeds (Figure 1), which could be also induced into the introgression lines during the backcrossing with bread wheat. Therefore, in addition to disease resistance genes, we will also focus on other important agricultural traits such as plant height and yield by using phenotypic and marker-assisted selection in the future backcrossing process.

## Figures and Tables

**Figure 1 pathogens-11-00025-f001:**
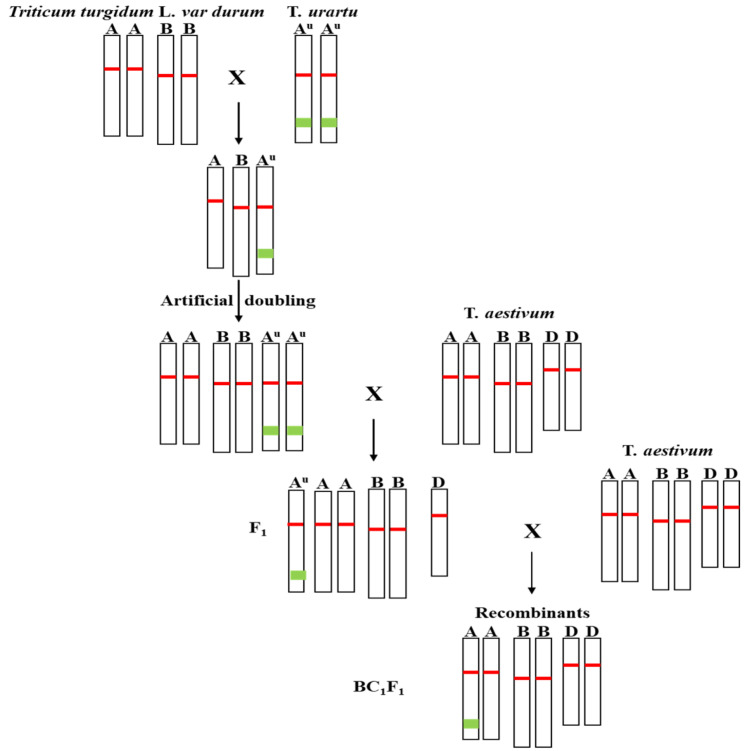
Workflow for the introduction of *Pm60* and *Pm60b* of chromosome 7A from *T. urautu* into common wheat. The green rectangle on the chromosome represents the *Pm60* or *Pm60b* region and the red rectangular box represents the centromere.

**Figure 2 pathogens-11-00025-f002:**
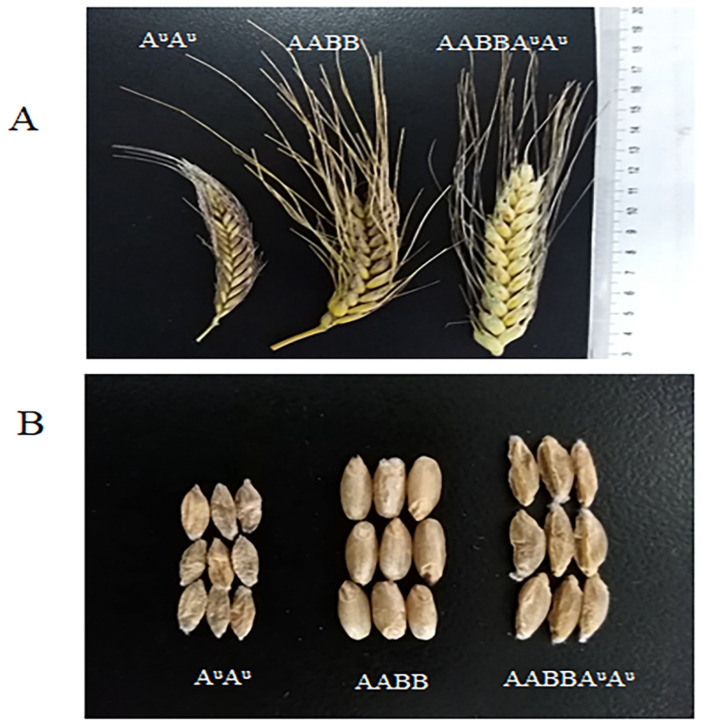
Variation in spike shape (**A**) and seed size (**B**) of different ploidy materials. A^u^A^u^ is *T. urartu* (PI428215), AABB is Mo75 and AABBA^u^A^u^ is the F_1_ material (Mo75 × PI428215) that the chromosome has been doubled by colchicine.

**Figure 3 pathogens-11-00025-f003:**
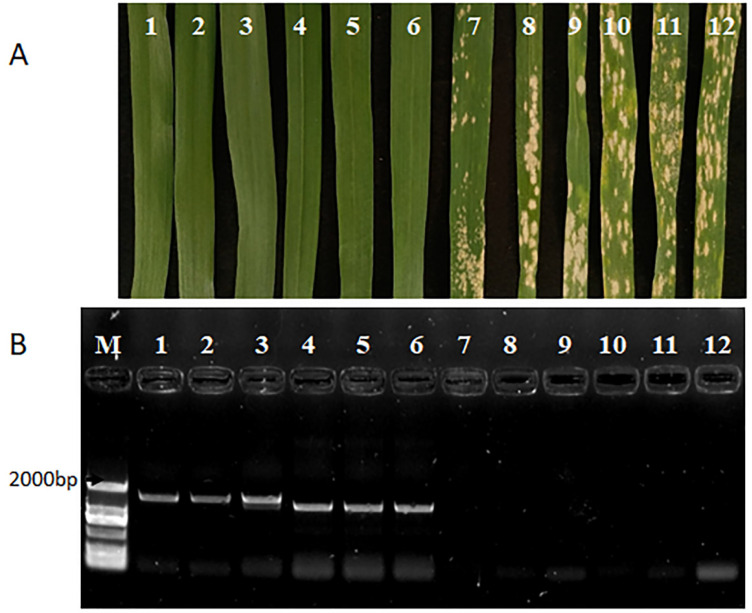
Marker-assisted screening of *Pm60* and *Pm60b* loci and the infection types of different introgression individuals. (**A**) The infection type (IT) of different introgression individuals at 13 days post-inoculation (1–6: IT = 0, 7–12: IT = 4). (**B**) Agarose gel electrophoresis of PCR products amplified with marker *M-Pm60* from 12 progenies; 1–3 are the resistant introgression individuals (Z1–1, Z5–3, Z10–1) containing *Pm60b*; 4–6 are the resistant introgression individuals (Z14–1, Z14–2 and Z14–6) containing *Pm60*; and 7–12 are the susceptible introgression individuals (Z1–2, Z5–1, Z10–2, Z10–3, Z14–4 and Z14–5) without functional *Pm60* alleles; M is marker (Direct-load StarMarker D2000 GenStar).

**Figure 4 pathogens-11-00025-f004:**
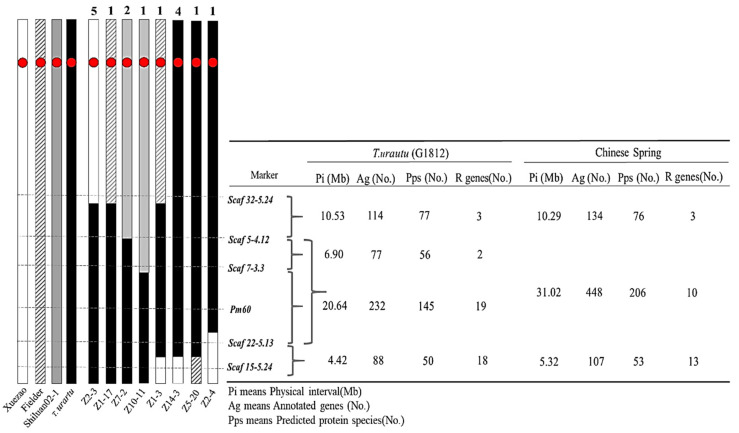
Recombination analysis using five *Pm60*-flanking markers. Left: Different types of recombinant plants based on marker analysis. The number of the recombinant types (above) and the name of representative introgression lines (below); Red circular boxes represent centromeres; right: physical location of the linked markers on G1812 and Chinese spring, as well as the number of genes annotated and the predicted protein types within the interval.

**Figure 5 pathogens-11-00025-f005:**
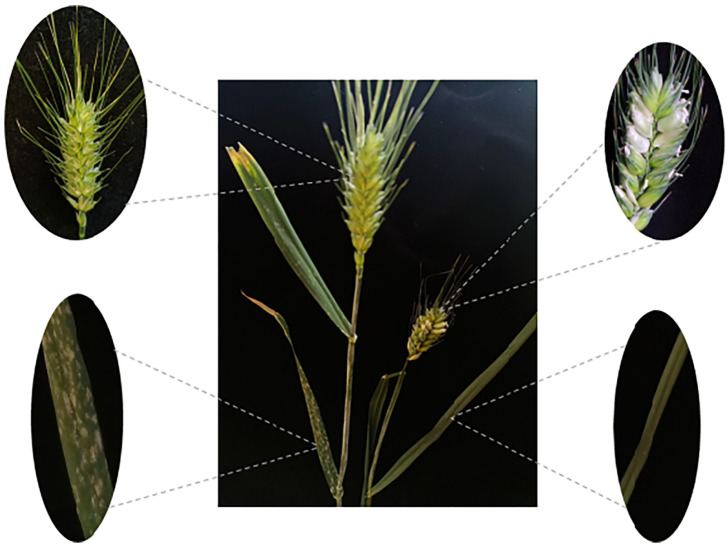
Resistant strains with a high selfing ability. Left: susceptible introgression individual (Z2–1) in the adult stage showing a lower selfing ability. Right: resistant individual (Z2–3) with a higher selfing ability.

**Table 1 pathogens-11-00025-t001:** The PCR primer sequences used in this study.

Marker	Primer Sequence (5′–3′)	Purpose
*M-Pm60 F* *M-Pm60 R*	CATTAACTTTGAGTTGTTGGA CGGTGATCATACCAGAATTC	Screening of the *Pm60* alleles
*scaf15–5.24 F* *scaf15–5.24 R*	CATCATCACAAAGATACCGAT TCGTCTTTATCTCTGCTCCT	Linkage marker with *Pm60* alleles
*scaf22–5.13 F* *scaf22–5.13 R*	AATGTAGATCGTCTCTTCGC ACACCTTGTGTTTTTGCTCT	Linkage marker with *Pm60* alleles
*scaf7–3.3 F* *scaf7–3.3 R*	CGGTTCCATCTCACATTTTG ATTGAGAGTTCGGGATTTGG	Linkage marker with *Pm60* alleles
*scaf5–4.12 F* *scaf5–4.12 R*	AGATGAAATTGAGCGAAGTT ATAATCAATGTCCACCGAAG	Linkage marker with *Pm60* alleles
*scaf32–5.24 F* *scaf32–5.24 R*	CCACCTCATGAACAACTACC AGGAAACACAACACAACAGG	Linkage marker with *Pm60* alleles

Note: Detailed information on these markers can be found in a previous report [26].

**Table 2 pathogens-11-00025-t002:** Segregation of powdery mildew resistance in six backcross populations (BC_1_F_1_).

No.	Male (♂)	Female (♀)	Number of Seedlings			
			Resistant	Susceptible	χ2 _(1:1)_	*p*-Value
Z1	Fielder	(Mo75/ PI428215)/Xuezao	17	23	0.40	0.53
Z2	Xuezao	(Mo75/ PI428215)/Xuezao	19	9	6.3	0.01
Z5	Fielder	(Mo75/ PI428315)/Xuezao	9	13	0.18	0.67
Z7	Shiluan 02–1	(Mo75/ PI428215)/Xuezao	17	15	0.51	0.48
Z10	Shiluan 02–1	(Mo75/ PI428315)/Xuezao	23	21	0.37	0.54
Z14	Xuezao	(Mo75/ CITR17664)/Xuezao	22	6	29.17	0.07 × 10^–5^

Note: Common wheat (AABBDD): Fielder, Xuezao, Shiluan02–1; Durum (AABB): Mo75; *T. urartu*: PI428215, PI428315, CITR17664.

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
