# Peer review of "Introgression of the Powdery Mildew Resistance Genes Pm60 and Pm60b from Triticum urartu to Common Wheat Using Durum as a ‘Bridge’"

_pathogens, 2021, doi:10.3390/pathogens11010025_

Round 1

Reviewer 1 Report

Using of genetic resistance can be the best way of crop protection, mainly those grown on large areas as wheat commonly is. Genes of specific resistance are often very attractive due to their high efficiency against pathogen isolates (response type close to 0) and simple inheritance. However, their effectiveness to the pathogen population is usually low and durability of the positive effect  in cultivars is short, especially when the crop is widely grown. It is my critical comment for introduction. Regards to a part of evaluation (Significance of Content) it is high from research point of view but rather low or even misleading for practical applications.

Because the working place of authors is located in the area of T. urartu natural occurrence and the institution is the leader for evolution of plants I would expect at least some information if wild accessions carrying studied alleles are resistant/susceptible in the nature, something about coevolution within the given plant pathosystem and information if the effectiveness of both alleles was measured somehow, at least with a set of pathogen isolates.

If the above mentioned introduction is omitted, than the contribution itself is good, based on original approach and carefully accomplished experiments and generally well interpreted results. I did not notice important flaws.

Some technical remarks

In keywords „Powdery mildew“ is listed. Instead I recommend authors to use „Wheat powdery mildew“. Perhaps also T. turgidum subsp. durum should be there.

What method for inoculation was used.

What does it mean „A sufficient amount of Bgt spores was used to infect two-week-old wheat plants.“

In the following sentence the numeral [26] should be transfer behind Zhao et al. “In the previous report, Zhao et al. found that T. urartu accessions CITR 17664 contained Pm60, PI 428215, and PI 428315 contained Pm60b [26].“

In the note below Table 1 The detailed information of these markers can be found in Zou et al., (2018)“ the order number of the citation is missing at all and comma is unnecessary there.

The Bgt isolate E09… was kindly provided by Prof. Xiayu Duan, Institute of Plant Protection, Chinese Academy of Agricultural Sciences should be removed from M&M to Acknowledgments.

Author Response

Thanks a lot for your comments!

Reviewer 2 Report

This manuscript described that the durum can be a ‘bridge’ for introducing the target gene, Pm60, from diploid T. urartu into the common wheat. The results have a high value for wheat resistant-breeding. Some comments are listed below:

1, In abstruct, ‘Moreover, we have selected two resistance-harboring introgression lines with high self-fertility.’ It is better the author can add the aim for mentioning these lines with high self-fertility, such as, breeding.

2, As the Pm60 has been cloned, the authors should describe more details/backgrounds in terms of Pm60 and its alleles in ‘Introduction’, and their resistant mechanism as well. e.g. What are differences among Pm60, Pm60a and Pm60b?

3, Hexaploid wheat (SHW, AABBAuAu) showed the unfavorable traits of seeds (Fig 2B), this may lead the yield losses, compare with Mo77. The authors at least should discuss this point and provide some potential strategies for next step in Discussion part.

4, Lines 165-166,' A powdery mildew test indicated that 165 these SHWs were highly resistant (IT=0-0;) to Bgt isolate E09 at all stages'. This sentence has no data to support, please provide the indication.

5, Z1-4 and Z1-10 showed high self-fertility, what are fertility rates of them, and how about normal fertility of other recombinant progenies?

Author Response

Thanks a lot for your comments!
